# The Complexity of Oxidative Stress in Human Age-Related Diseases—A Review

**DOI:** 10.3390/metabo15070479

**Published:** 2025-07-15

**Authors:** Alicja Płóciniczak, Ewelina Bukowska-Olech, Ewa Wysocka

**Affiliations:** Department of Laboratory Diagnostics, Poznan University of Medical Sciences, 84 Szamarzewskiego Str., 60-569 Poznań, Polandewysocka@ump.edu.pl (E.W.)

**Keywords:** oxidative stress, antioxidant activity, reactive oxygen species, age-related disorders, genetic predisposition

## Abstract

The aging process is a complex and dynamic phenomenon influenced by genetic, environmental, and biochemical factors. One of the key contributors to aging and age-related diseases is oxidative stress, resulting from an imbalance between the generation of reactive oxygen species (ROS) and the efficiency of antioxidant defense systems. In this review, we introduce the concept of the oxidative stress complexity—a network encompassing ROS-generating systems, enzymatic and non-enzymatic antioxidants, and genetic determinants that collectively shape redox homeostasis. Emerging research highlights the significant influence of genetic variability on the activity and expression of selected and most examined antioxidant enzymes, including superoxide dismutase (SOD), paraoxonase 1 (PON1), catalase (CAT), and glutathione peroxidase (GPX), thereby modulating individual susceptibility to oxidative damage, disease onset, and the pace of aging. Particular attention is paid to the interplay among oxidative stress, chronic inflammation, and metabolic dysfunction in the pathogenesis of various age-related disorders. By integrating findings from molecular studies, clinical research, and population genetics, we discuss the diagnostic and prognostic potential of antioxidant enzymes as biomarkers of aging and explore strategies for redox-modulating interventions. Understanding these interrelations is essential for identifying biomarkers of biological aging and developing personalized strategies aimed at promoting healthy aging and reducing the risk of chronic disease.

## 1. Introduction

Population aging is a well-documented global phenomenon with profound implications for healthcare systems. According to projections from the World Health Organization (WHO), by 2050, the number of individuals aged ≥60 years will exceed 1 billion, representing approximately 21% of the global population [1,2,3]. This demographic transition significantly impacts public health and healthcare infrastructure, socioeconomic policies, and financial sustainability, as aging is the primary risk factor for numerous chronic diseases, including cardiovascular disorders, type 2 diabetes mellitus, sarcopenia, neurodegenerative disorders such as Alzheimer’s disease, cancers, and frailty syndrome [4,5,6]. This matters particularly in countries where the aging population is increasing at an accelerated rate.

At the cellular level, aging is characterized by a progressive decline in physiological resilience, beginning after the third decade of life. It is associated with a gradual impairment in cellular and systemic homeostasis, resulting in systemic functional deterioration involving multiple organs [7,8]. A summary of selected age-related changes is shown in Table 1.

Although significant research has been dedicated to elucidating the molecular mechanisms of aging and identifying reliable biomarkers of senescence, the pathophysiology of aging requires more advanced studies. Many age-related pathologies share common risk factors, including physical inactivity, poor dietary habits, obesity, and genetic predispositions affecting the lipid and glucose metabolisms [21,22,23].

However, among the key contributors to these changes is oxidative stress, a condition resulting from an imbalance between the production of reactive oxygen species (ROS) and the efficiency of antioxidant defense mechanisms [12]. Oxidative stress has been implicated not only in the normal aging process but also in the pathogenesis of multiple age-related diseases. Within this conceptual framework, the free radical theory of aging posits that lifelong oxidative damage accrues in tandem with a gradual disruption of pro-oxidant/antioxidant homeostasis—an axis that constitutes a central focus of the present review—and the corpus of work initiated by early pioneers such as D. Harman in the 1950s and further advanced by W. Pryor, R.G. Cutler, J.A. Knight, and R.S. Sohal continues to inspire current preventive medicine strategies aimed at mitigating oxidative stress-mediated, age-related diseases [24].

As the multifactorial background of age-related diseases is still under discussion, oxidative stress is one of the proposed linking mechanisms.

There is a lack of comprehensive reviews that integrate antioxidant enzyme systems with genetic variability and their clinical relevance in elderly populations. Prior works often examine enzymatic pathways or gene polymorphisms separately, and rarely explore their interaction in the context of population-based data or therapeutic potential.

A more comprehensive understanding of these mechanisms may provide insights into novel therapeutic strategies to prolong health span and mitigate age-associated disease burden. Therefore, the aim of this review is to explore the interplay between enzymatic antioxidant system and its genetic variability in the context of aging and age-related diseases.

## 2. Oxidative Stress

Oxidative stress arises from an imbalance between the production of reactive oxygen species (ROS) and the capacity of antioxidant systems to neutralize them. While ROS—including superoxide anions (O_2_^•−^), hydroxyl radicals (^•^OH), and hydrogen peroxide (H_2_O_2_)—play physiological roles in signaling and immune defense, their excessive accumulation damages lipids, proteins, and DNA, thereby accelerating cellular senescence [5] and may cause endothelial dysfunction and vascular remodeling [23].

In aging organisms, mitochondrial dysfunction becomes a main source of ROS, particularly due to inefficiencies in the electron transport chain (ETC). This is exacerbated by the decline in mitochondrial quality control, leading to the increased leakage of electrons and superoxide formation. Other significant sources include NADPH oxidases (NOX) [13], peroxisomal β-oxidation, and xanthine oxidase activity during purine metabolism.

This oxidative–antioxidant imbalance is implicated in numerous age-associated pathologies, such as diabetes mellitus, CVD, and neurodegenerative disorders [25]. What is more, it is also involved in insulin resistance, hormonal changes, and inflammaging and is described as an underlying cause of metabolic and frailty syndrome and sarcopenia [26,27,28,29].

Pathologically increased ROS levels facilitate uncontrolled cellular proliferation, apoptosis, and the exacerbation of inflammatory cascades. Persistent, low-grade chronic inflammation, known as “inflammaging,” results from the continuous secretion of pro-inflammatory cytokines by senescent cells, propagating a deleterious cycle of ROS accumulation, extracellular matrix degradation, apoptotic cell death, and tissue necrosis [30,31]. The necrotic cells and damaged extracellular matrix (ECM) generate signals that further amplify inflammation, creating a cycle of excessive free radical production and oxidative stress [32,33,34,35].

ROS overproduction initiates a cascade of deleterious events, such as:−Chronic low-grade inflammation (inflammaging) via the activation of NF-κB and release of pro-inflammatory cytokines [19,36,37];−Endothelial dysfunction, marked by nitric oxide depletion and vascular stiffening [38,39];−Impaired insulin signaling and increased oxidative burden in adipose tissue [40];−And DNA damage, contributing to telomere attrition and apoptosis.

Maintaining equilibrium between ROS generation and antioxidant defense is crucial for cellular homeostasis. Dysregulated ROS levels and subsequent oxidative stress significantly influence aging mechanisms and accelerate the progression of age-related diseases. Increasing evidence suggests that oxidative stress plays a pivotal role in aging processes and the progression of age-related diseases [41]. Understanding the interplay between ROS production, antioxidant defense, and redox signaling is essential for unraveling the pathophysiology of aging and associated diseases, and provides a foundation for redox-based therapeutic strategies [42].

## 3. Antioxidant Mechanisms

As mentioned before, to mitigate the effects of free radicals, the human body presents antioxidants (both enzymatic and non-enzymatic), which play a crucial role in maintaining redox balance, essential for proper physiological function, especially in the context of aging, where antioxidant capacity progressively declines.

Oxidative stress arises when antioxidant levels are insufficient, leading to disruptions in redox homeostasis and an accumulation of free radicals.

Among the enzymatic antioxidants, superoxide dismutase 1 (SOD1), glutathione peroxidase (GPX), catalase (CAT), and thioredoxin (Trx) constitute the primary defense lines. SOD1 is a key intracellular antioxidant enzyme that catalyzes the conversion of superoxide anions (O_2_^•−^) into oxygen (O_2_) and hydrogen peroxide (H_2_O_2_). By regulating superoxide levels, SOD1 plays a crucial role in cellular metabolism, immune defense, and oxidative stress response. However, its expression and function can be limited by different metabolic and environmental factors, influencing hydroxyl radical formation [43]. Its expression is regulated by transcription factors, including NF-κB, SP1, AP-1, AP-2, and C/EBP, which respond to oxidative stress and cytokine signaling [44]. Notably, aging and chronic inflammation alter the regulation of these pathways, leading to impaired antioxidant responses.

In parallel, mitochondrial antioxidant enzymes, such as SOD2, peroxiredoxin 3 (Prx3), and Trx2, are regulated by PGC-1α, a master transcriptional coactivator that becomes downregulated during aging. This contributes to mitochondrial dysfunction and increased ROS leakage, a hallmark of cellular senescence.

The clinical significance of SOD1 was first established in neurology [45]. Given its central role in oxidative stress regulation, SOD1 remains a critical focus in aging, neurodegeneration, and metabolic disorders research.

GPX facilitates the reduction of H_2_O_2_ and lipid hydroperoxides (LOOH), while CAT converts H_2_O_2_ into water (H_2_O) and molecular oxygen (O_2_). Thioredoxin (Trx) also contributes to H_2_O_2_ detoxification by facilitating its reduction to water [46]. These enzymes also regulate transition metal availability, inhibiting Fenton reactions that generate highly reactive ROS. Studies suggest a decline in antioxidant enzyme activity with aging, as it is associated with a decline in peroxisome proliferator-activated receptor γ coactivator-1α (PGC-1α) expression, which regulates mitochondrial antioxidant enzymes such as superoxide dismutase-2 (SOD-2), CAT, peroxiredoxins (Prx3 and Prx5), and thioredoxin. Similarly, impaired cytosolic antioxidant enzymes, including SOD1, GPX, and heme oxygenase-1 (HO-1), contribute to increased oxidative stress [47].

Non-enzymatic antioxidants are classified into lipophilic and hydrophilic compounds. Lipophilic antioxidants include carotenoids, ubiquinol, and α-tocopherol, while hydrophilic antioxidants comprise vitamin C, bilirubin, uric acid, albumin, and flavonoids [48]. Glutathione, vitamins C and E, carotenoids, and uric acid play a supportive role in scavenging free radicals and stabilizing membrane integrity. However, enzymatic antioxidants remain the primary system for ROS detoxification under physiological and pathological conditions.

## 4. Age-Related Metabolic Challenges for the Antioxidant System

Aging is associated with significant alterations in the endocrine system, including increased glucocorticoid levels, which are known to contribute to inflammatory processes [49]. Concurrently, the concentrations of sex hormones decline, with reduced testosterone in men and estradiol in women [20]. Additionally, the secretion of growth hormone (GH) decreases, reducing insulin-like growth factor 1 (IGF-1) synthesis, which negatively impacts muscle mass and function [19]. IGF-1 also plays a protective role against oxidative stress and serves as a marker of frailty status in the elderly [50].

Moreover, the combination of increased inflammation, elevated reactive oxygen species production, and impaired antioxidant defense mechanisms disrupts insulin synthesis, secretion, and action, ultimately leading to insulin resistance [51,52]. The progressive decline in insulin sensitivity is a primary driver of hyperglycemia, which serves as a precursor to microangiopathies affecting the ocular, renal, and nervous systems, as well as atherosclerosis, contributing to cardiovascular complications [53,54].

Beyond its role in glucose metabolism, insulin resistance also influences lipid metabolism and protein synthesis. The pathophysiology of this process is closely linked to oxidative stress and chronic low-grade inflammation in the adipose tissue, where adipocytes and macrophages produce ROS and pro-inflammatory adipocytokines, further exacerbating insulin resistance. These immunometabolic alterations in the adipose tissue are central to the development of systemic inflammation and oxidative stress, which contribute to the progression of age-related complications and comorbidities [55,56]. Aging also correlates with vascular dysfunction and atherosclerosis, where endothelial nitric oxide (NO) bioavailability declines due to reduced endothelial nitric oxide synthase (eNOS) activity, while pro-oxidant NADPH oxidase enzymes become overactive [57]. The NADPH oxidase system, a major contributor to oxidative stress, is upregulated in endothelial cells, driving chronic inflammation and atherosclerotic plaque progression [58]. Inflammatory signals further suppress eNOS activity, exacerbating endothelial dysfunction [59].

## 5. Antioxidant Enzyme Activities

The status of antioxidant enzymes—whether increased, optimal, or decreased—can reflect either the consequences of reactive oxygen species (ROS) accumulation or potential genetic variations in enzymes. Valuable insights can be gained from studies that simultaneously assess both the activity levels of antioxidant enzymes and the concentrations of ROS reaction products. Modern laboratory techniques, particularly in molecular biology, provide additional tools to clarify the underlying mechanisms of altered enzymatic activity. At the same time, it is important to recognize the complexity of the antioxidant defense system, where individual enzymes are responsible for specific aspects of cellular protection against oxidative stress.

The proper function of this system depends on the coordinated activity of multiple components, including various antioxidant enzymes.

### 5.1. SOD

Among the enzymatic antioxidants, superoxide dismutase (SOD) initiates the inactivation reactions of reactive oxygen species, converting the superoxide anion radical into the less active hydrogen peroxide. This enzyme protects the cell against ^•^O_2_^−^ and indirectly prevents the formation of the ^•^OH radical in the Fenton reaction catalyzed by transition metals (Fe^2+^, Cu^2+^). Subsequent reactions limiting the reactivity of oxygen involve, among others, catalase and glutathione peroxidase, catalyzing the decomposition of hydrogen peroxide [60,61].

The clinical significance of superoxide dismutase was initiated by neurology. Amyotrophic lateral sclerosis (ALS), especially the familial form, is the disease in which superoxide dismutase activity as well as mutations and the expression of the SOD1 gene have been best studied, yet the precise mechanism by which each of the described mutations leads to neurodegenerative disease remains unclear [62]. Mutations with different properties produce the same clinical effect and in laboratory tests—e.g., reduced Cu and Zn-SOD (SOD1) activity in erythrocytes [63]. Based on SOD, we can illustrate the efforts of researchers to explain oxidative stress in humans—selected examples are presented below.

Transcription factors involved in the regulation of SOD1 expression, both constitutive and induced [64], include:−Nuclear Factor-KappaB (NF-κB), sensitive to changes in the redox state in the cell [65];−Specificity Protein 1 (SP1 protein) [66];−Activator Protein-1 (AP-1), sensitive to, among others, cytokines and oxidative stress, described in the processes of cell proliferation and neoplastic transformation; in relation to SOD1, it is supposed to reduce gene transcription [67];−Activator Protein-2 (AP-2)—family of proteins through which ginsenoside Rb2 (active substance from the root of Panax ginseng) can increase sod1 transcription [68];−Proteins binding to the regulatory and enhancing sequence CCAAT, the so-called C/EBP (CCAAT-Enhancer-Binding Proteins), necessary for the basic transcription of the SOD1 [69].

Researchers also suggest special transcriptional interactions described in animal studies, e.g., positive, via the transcription factor Elk1, and negative, via YY1 [70], and that arachidonic acid activates SOD1 via the peroxisome proliferator response element (PPRE) [71]. On the other hand, the anticancer drug mitomycin C inhibits the transcription of SOD1 in humans via the p-53 factor [72], and some xenobiotics (e.g., 2,3,7,8-tetrachlorodibenzo-p-dioxin, TCDD) can stimulate the induction of SOD1 expression by acting on the ARE (antioxidant responsive element) or XRE (xenobiotic responsive element) areas [73].

The end result, which is enzyme activity, is not a simple translation of gene structure into the amount of enzyme protein and its activity. In recent years, we have come to appreciate epigenetic regulation, which concerns changes in gene expression without changes in DNA sequence—covalent DNA modifications and changes in chromatin structure. Epigenetics may explain phenotypic differences resulting from different gene expression of genetically identical cells [74].

SOD1 is a key antioxidant enzyme that regulates the level of superoxide anion radicals in the cytosol, which are necessary for normal cellular metabolism (including the regulation of signaling pathways and defense against pathogens), but also appear during pathological processes and under external factors, limiting the generation of hydroxyl radicals from •O_2_^−^ [60].

Significant Cu and Zn deficiencies in people fed orally (as opposed to those receiving long-term parenteral nutrition) are relatively rare; however, reduced Zn concentration and SOD1 activity as well as increased thiobarbituric acid reactive substance (TBARS) concentration were observed in the erythrocytes of obese men, with unchanged concentrations of copper and iron in the cell, compared to the group of healthy men with a normal body mass index (BMI) [75]. In studies assessing the oxidative–antioxidant balance in basic fasting conditions, researchers observed reduced SOD activity and increased TBARS concentration in the plasma of people with type 1 and type 2 diabetes, compared to the normoglycemic group [76]. Colak et al. documented significantly reduced erythrocyte SOD1 activity and plasma total antioxidative status (TAS) in patients with type 2 diabetes and CVD compared to healthy individuals [77]. The relationship between diabetes and CVD makes the analysis of the mechanisms proposed as common to both pathologies particularly important. According to the authors of a community-based cohort study, the increased plasma SOD activity was associated with a reduced all-cause mortality among older women [78].

Although extensively studied in the context of familial ALS, SOD1’s role in age-related neurodegeneration may differ mechanistically from inherited mutations. Nevertheless, insights from ALS models have contributed to understanding SOD1 regulation, cofactor dependence (Cu/Zn), and stress-induced transcriptional responses.

### 5.2. CAT and GPX

Catalase (CAT) and glutathione peroxidases (GPX) are essential components of the cellular antioxidant defense system, primarily responsible for the detoxification of hydrogen peroxide (H_2_O_2_)—a reactive oxygen species produced both physiologically and during oxidative stress. While CAT catalyzes the direct dismutation of H_2_O_2_ into water and oxygen, GPX reduces H_2_O_2_ and lipid hydroperoxides using reduced glutathione (GSH) as a cofactor, contributing to the maintenance of redox homeostasis [54].

During aging, a decline in CAT and GPX activity has been observed and is thought to reflect cumulative oxidative damage to peroxisomes and mitochondria, as well as the age-related dysregulation of transcriptional control. PGC-1α, a master regulator of mitochondrial antioxidant enzymes, including CAT and GPX 1, shows decreased expression with aging, contributing to reduced enzymatic activity and enhanced ROS accumulation [31]. Moreover, chronic inflammation and hormonal dysregulation common in elderly individuals exacerbate this decline by altering enzyme expression and cofactor availability.

Clinical and observational studies support the link between reduced CAT/GPX activity and oxidative stress-related conditions in aging populations. For example, in older patients with metabolic syndrome, a high plasma CAT activity was interpreted as a compensatory response to persistent oxidative stress, particularly among those with stable or elevated BMI [72]. Similarly, increased GPX activity following dietary interventions in post-acute coronary syndrome (ACS) patients suggests the potential reversibility of age-related redox imbalance under favorable metabolic conditions [79].

Importantly, a decreased GPX activity has also been reported in elderly individuals with type 2 diabetes and cardiovascular comorbidities, indicating a possible mechanistic link between antioxidant enzyme dysfunction and disease burden [70]. However, inconsistencies in study outcomes may reflect interindividual variability in gene expression, nutritional status, or underlying inflammatory tone.

### 5.3. PON1

Paraoxonase 1 (PON1) is an enzyme secreted by the liver and primarily associated with high-density lipoproteins (HDLs) in the circulating blood, with a smaller fraction bound to other lipoproteins. PON1 exhibits a remarkably broad substrate specificity and exerts three distinct hydrolytic activities: (1) paraoxonase activity, which enables the hydrolysis of toxic organophosphates such as paraoxon, the active oxon metabolite of the insecticide parathion; (2) arylesterase activity, facilitating the breakdown of non-phosphorous arylesters; and (3) lactonase activity, responsible for hydrolyzing a variety of lactones, including lipid-derived lactones formed during fatty acid oxidation.

Although paraoxonase 1 is not a classical antioxidant enzyme that directly scavenges free radicals, it exerts a significant protective effect against oxidative stress through indirect mechanisms. PON1 hydrolyzes lipid peroxides, especially those formed during the oxidation of low-density lipoprotein (LDL) particles, thereby preventing the accumulation of oxidized LDL (oxLDL) and inhibiting subsequent pro-atherogenic processes [46,47]. This activity contributes to the preservation of endothelial integrity and reduces the inflammatory activation of vascular cells. Additionally, PON1 plays a central role in maintaining the antioxidative function of high-density lipoprotein, enhancing its role in reverse cholesterol transport and reducing foam cell formation [48,49]. Importantly, PON1 exhibits lactonase activity, which enables it to degrade reactive lipid lactone cytotoxic byproducts that accumulate during oxidative stress—thereby further reducing redox imbalance [50]. Genetic common variants that impair *PON1* expression or enzymatic efficiency have been linked to increased oxidative burden, endothelial dysfunction, and a greater susceptibility to age-related conditions such as cardiovascular disease and metabolic syndrome [51,52]. The results of research on antioxidant enzymes in the aging population are summarized in Table 2.

An interesting study was conducted on a population aged 18–114 years by Erdman et al., where different antioxidant genes (*PON1, PON2, MTHFR, MSRA, SOD1, NQO1, and CAT*) were investigated. In this 20-year follow-up study in the Volga-Ural region, the protective common variants *PON1*, i.e., rs662 was found to be linked to longevity [84].

## 6. Genetic Predisposition to Oxidative Stress Disturbances

The ability of oxidative stress management has been linked to genetic variations, commonly found in populations, known as single nucleotide variants (SNVs). In the medical literature, these are often referred to as polymorphisms. These genetic alterations can lead to either decreased antioxidant activity or increased oxidative stress, which in turn may contribute to aging and various age-related conditions such as CVD, type 2 diabetes mellitus, atherosclerosis, chronic kidney disease (CKD), retinal or macular degeneration, among others [85].

Oxidative stress defense relies on key antioxidant enzymes such as superoxide dismutases encoded by the *SOD1* (Mendelian Inheritance in Man; MIM: 147450), *SOD2* (MIM: 147460), and *SOD3* (MIM: 185490) genes; catalase encoded by the *CAT* gene (MIM: 115500); glutathione peroxidases encoded by the *GPX1* (MIM: 138320), *GPX2* (MIM: 138321), and *GPX4* (MIM: 138322) genes; and paraoxonase encoded by the *PON1* (MIM: 168820), *PON2* (MIM: 602447), and *PON3* (MIM: 602720) genes. To date, SOD1 and its interactions with other proteins have been the most extensively examined by researchers (Figure 1).

This figure presents the protein–protein interaction network of SOD1 and its associated partners. The diagram highlights high-confidence interactions, with scores ranging from 0.999 to 0.964, suggesting strong functional associations between SOD1 and other proteins. Among the most prominent interacting partners are CCS, PARK7, VDAC1, SOD2, FUS, TARDBP, NEFL, HSPA5, DERL1, and BCL2. These proteins are mainly involved in processes such as oxidative stress response, protein folding, mitochondrial function, and neurodegeneration. From the network, it is clear that SOD1 acts as a central hub, interacting with numerous proteins, which underlines its role in maintaining cellular balance. The strongest interaction is with CCS, a copper chaperone that directly contributes to SOD1’s proper maturation and activity. Proteins like PARK7 and SOD2, known for their roles in protecting mitochondria and regulating oxidative stress, suggest that SOD1 may be tightly linked to mitochondrial function. Furthermore, connections with FUS, TARDBP, and NEFL, all linked to amyotrophic lateral sclerosis (ALS), highlight SOD1’s involvement in neurodegenerative disease pathways. Its interaction with HSPA5 and DERL1 points to a role in the unfolded protein response and ER-associated degradation, indicating a broader role in protein quality control. Finally, the presence of BCL2 suggests a potential link between SOD1 and apoptosis regulation. This network emphasizes the nature of SOD1 and supports its involvement in key cellular pathways related to oxidative stress and proteostasis.

Pro-oxidant genes include *XDH* (MIM: 607633), *CYBA* (MIM: 608508), *CYP1A1* (MIM: 109330), *PTGS2* (600262), *NOS1* (MIM: 163731), *NOS2* (MIM: 163730), *NOS3* (MIM: 163729), and *MAO* (MIM: 309860) [87]. Changes in both antioxidant and pro-oxidant genes can increase cellular oxidative stress, potentially contributing to oxidative stress-related diseases, including those linked to aging [85]. A summary of key oxidative stress-related genes associated to age-related disorders is shown in Table 3.

Oxidative stress has been shown to influence a few signaling pathways responsible for cellular defense and the restoration of the redox state balance. First, the Nrf2/ARE (nuclear factor erythroid 2-related factor 2/antioxidant response element) pathway is activated in response to oxidative stress, triggering the expression of over 200 genes involved in inflammation and oxidative stress defense. Second, the NF-κB pathway upregulates the expression of antioxidative genes, including *SOD2* and *GPX4*. Next, the PI3K/AKT pathway promotes the production of nitric oxide through the phosphorylation of endothelial nitric oxidative stress synthase (eNOS), leading to vascular tone modulation (Figure 2). Finally, the ferrotopic, apoptotic, FoxO, and ErB pathways regulate cellular responses to oxidative stress, including the induction of survival mechanisms. Moreover, they also activate the Nrf2 pathway, further enhancing cellular defense [85,111].

### 6.1. Cardiovascular Disorders

Researchers have shown that disturbances in oxidative stress mechanisms increases susceptibility to cardiac disorders [112]. PON1 protects against cardiovascular disease by preventing LDL oxidation, while the variant NM_000446.7 (*PON1*):c.575A>G p.Gln192Arg (rs662) affects enzyme activity, with the G allele linked to altered paraoxonase function and potential cardiovascular risk [113].

Similarly, myeloperoxidase (MPO), which is involved in immune defense, contributes to LDL oxidation, as the 5′UTR variant, i.e., NM_000250.2 (*MPO*):c.-463G>A (rs2333227), potentially increases its expression. Next, the variant of the *SOD1* gene NM_000636.4 (*SOD2*):c.47T>C p.Val16Ala (rs4880) disrupts manganese superoxide dismutase (MnSOD) transport, required for neutralizing reactive oxygen species (ROS) within mitochondria, and results in higher cellular oxidative stress. Nitric oxide, produced by endothelial nitric oxide synthase (eNOS), supports vascular function; however, eNOS variants like NM_000603.5 (*NOS3*):c.894G>T p.Glu298Asp (rs1799983) and c.-786T>C (rs2070744) impair enzyme activity, reducing nitric oxide levels and affecting blood vessel dilation. As a consequence, patients harboring *NOS* c.894G>T p.Glu298Asp alteration present a higher incidence of myocardial infarction and carotid arteriosclerosis [114].

This variant was also described as a risk factor of hypertension, particularly in Caucasians and in North African populations. Additionally, two variants of the *CYBA* gene, i.e., NM_000101.4 (*CYBA*):c.214T>C p.His72Tyr (rs4673) and *CYBA* c.-932A>G (rs9932581), are associated with heightened oxidative stress and an increased risk of atherosclerosis as *CYBA* encodes the alpha subunit of cytochrome b(-245), which is a component of the NADPH oxidase (NOX) complex, required for ROS generation [102].

### 6.2. Type 2 Diabetes Mellitus

Some changes in antioxidant genes have been linked to both the etiology and vascular complications of type 2 diabetes mellitus. In study from Saudi Arabia, two of the investigated SNVs were relevant, i.e., rs17856199 in the *GSTT1* gene and rs2297518 in the *NOS2* gene, and showed a significant association with disease risk. The study encompassed 384 unrelated adult individuals, i.e., 177 with type 2 diabetes mellitus and 207 healthy controls [115].

In addition, a comprehensive meta-analysis of 73 studies, including over 130,000 individuals, pointed to the protective effect of the p.Pro12Ala variant (rs1801282) in the *PPARG* gene, particularly in European, East Asian, and Southeast Asian populations [116]. The *PPARG*, Peroxisome Proliferator-Activated Receptor Gamma, gene is involved in glucose metabolism, lipid homeostasis, and inflammation and acts as a transcription factor that regulates mitochondrial function and redox balance. It increases the expression of superoxide dismutase (SOD) and catalase enzymes, and downregulates pro-oxidant NADPH oxidase [117].

### 6.3. Cancer

Oxidative stress plays a dual role in cellular homeostasis, acting as both a protective mechanism and a cause of tumor progression. It has been shown that oxidative stress induces the Nrf2 pathway, which activates not only antioxidant genes but also several oncogenes such as *MMP9, BCL-2, BCL-xL, TNF-α*, and *VEGF-A*, leading to tumor survival and angiogenesis [118]. In addition, elevated ROS levels can stimulate key oncogenic signaling pathways, such as PI3K/AKT and MAPK, which are involved in cell proliferation, survival, and migration. The activation of these pathways enhances epithelial–mesenchymal transition, invasion, and metastasis [119].

In a large cohort study of 703 Danish breast cancer–control pairs, no correlation between the tested variants of the *SOD1* (rs202445), *CAT* (rs1001179, rs769217, rs12270780), and *GSR* (rs1002149) genes and breast cancer were shown. However, the intake of alcohol, fruit and vegetables, and smoking status interacted with some of the tested variants in relation to breast cancer risk [88]. On the other hand, it has been described that *SNRPB* expression levels, encoding small nuclear ribonucleoprotein-associated proteins B and B’, promote oxidative stress and ferroptosis in hepatocellular carcinoma [120].

### 6.4. Accelerated Aging Diseases

Genetic diseases accompanied by premature aging include Hutchinson–Gilford Progeria Syndrome (MIM: 176670), Werner syndrome (MIM: 277770), and Cockayne syndrome A (MIM: 216400) and B (MIM: 133540) [121]. Notably, oxidative stress imbalance has been stated in all of these conditions. Hutchinson–Gilford Progeria Syndrome results from *LMNA* mutations, leading to the synthesis of a mutant prelamin A, i.e., farnesylated prelamin A [122]. Its accumulation leads to ROS overproduction and, subsequently, impairs antioxidant defense mechanisms. In addition, the Nrf2 pathway repression weakens cellular resilience against oxidative damage [123,124]. Werner syndrome encompasses early ageing, excess cancer risk, high incidence of type 2 diabetes mellitus, early atherosclerosis, ocular cataracts, and osteoporosis. A possible involvement of redox pathways in Werner syndrome was shown via NAD^+^ depletion in affected individuals, as NAD^+^ regulates stress response, DNA damage repair, and energy metabolism [125]. In Cockayne syndrome A and B, studies on primary fibroblasts exposed to H_2_O_2_ have shown both the downregulation of DNA repair and cell cycle genes as well as telomere attrition rates, suggesting a potential involvement in the pathology of these syndrome through impaired response to oxidative stress and telomere instability [126].

## 7. Conclusions

Aging, metabolic disorders, and inflammatory processes impair antioxidant enzyme activity, contributing to elevated oxidative stress and an increased risk of chronic diseases such as type 2 diabetes, CAD, cancer, and neurodegenerative disorders. Altered hormone levels, increased proinflammatory cytokines, and dysregulated lipid metabolism further exacerbate redox imbalance. Oxidative stress is proposed to be an important underlying cause of these conditions. Enzymes such as SOD1 and PON1 may demonstrate diagnostic and prognostic value in age-related diseases and potential as therapeutic targets.

Genetic predisposition is considered to be one of the factors that may influence the efficiency of antioxidant defense systems and the body’s response to oxidative stress, a major contributor to aging and age-related diseases. However, the coexistence of aging and multimorbidity makes it extremely difficult to distinguish between genetic predisposition to senescence and age-related multimorbidity.

Understanding the dynamic interplay between antioxidant defenses, metabolic regulation, and inflammatory signaling might be crucial for identifying biomarkers of aging and designing redox-modulating interventions. Such strategies may offer promising avenues for mitigating oxidative damage and improving health span in aging populations.

The reviewed literature does not determine the superiority of genetics over biochemistry or vice-versa, but we hope it can inspire scientists to conduct more complex studies.

Single-marker approaches cannot capture oxidative stress’s complexity; multi-enzyme panels measured in matched bio-fluids are required.

Antioxidant therapies might be stratified by both enzymes’ genotype and enzymes’ activities, accompanying by the dominant ROS source.

It remains a scientific priority to design longitudinal studies that combine enzymatic activity profiling, genetic variability, and redox signaling data—while considering comorbidities commonly present in ageing populations—in order to define actionable thresholds of oxidative stress and to characterize what constitutes an optimal antioxidant status.

## Figures and Tables

**Figure 1 metabolites-15-00479-f001:**
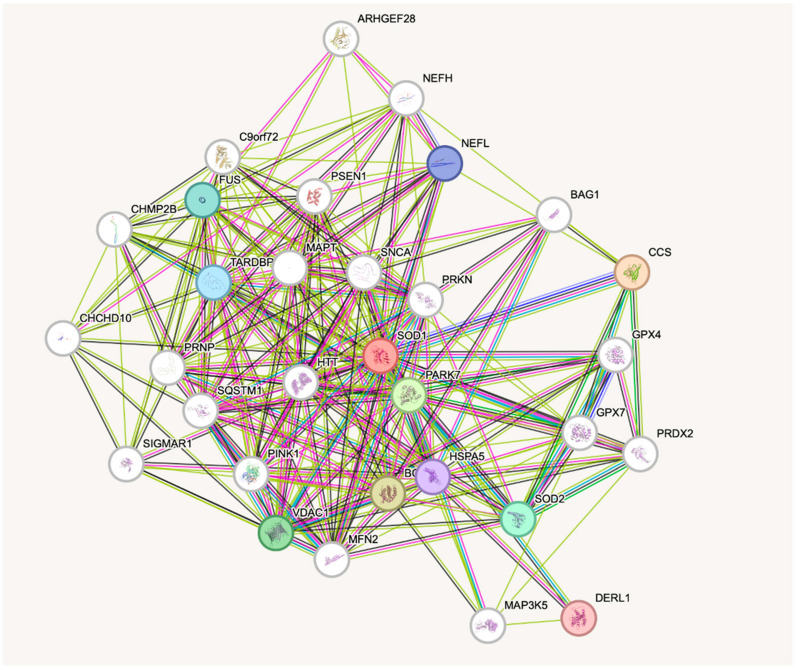
SOD1—possible interactions with other proteins—based on STRING database [86].

**Figure 2 metabolites-15-00479-f002:**
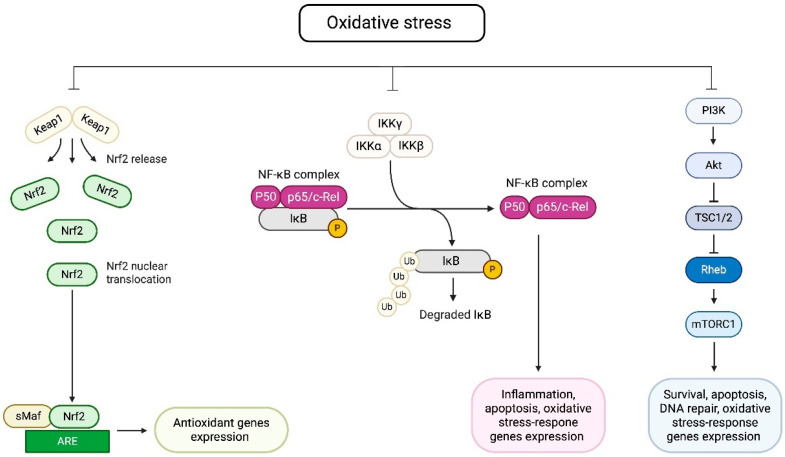
Cellular signaling pathways activated by oxidative stress. On the left—Oxidative stress disrupts the interaction between nuclear factor erythroid 2–related factor 2 (Nrf2) and Kelch-like ECH-associated protein 1 (Keap1), resulting in the release and nuclear translocation of Nrf2. Nrf2 forms a heterodimer with small Maf proteins (sMaf) and binds to antioxidant response elements (AREs), leading to the transcription of antioxidant genes. In the middle—Oxidative stress activates the IκB kinase (IKK) complex, i.e., IKKα, IKKβ, and IKKγ, which phosphorylates (marked as P) and promotes the ubiquitination (marked as Ub) of IκB. This targets IκB for degradation, allowing the NF-κB complex (p50 and p65/c-Rel) to translocate to the nucleus and drive the expression of genes involved in inflammation, apoptosis, and the oxidative stress response. On the right—The phosphoinositide 3-kinase-Protein Kinase B (PI3K-Akt) signaling pathway is activated by oxidative stress, leading to the downstream stimulation of the mechanistic target of rapamycin complex 1 (mTORC1) through tuberous sclerosis complex 1 and 2 (TSC1/2) and Ras homolog enriched in the brain (Rheb). This pathway supports cell survival, apoptosis regulation, DNA repair, and expression of oxidative stress-responsive genes. Figure created with BioRender.com and published with permission.

**Table 1 metabolites-15-00479-t001:** Selected physiological human body age-related changes. The authors’ compilation based on [9,10,11,12,13,14,15,16,17,18,19,20].

Human Body System	Age-Related Changes in Laboratory Tests
Digestive system	Liver dysfunction: increase in globulin, VII and VIII factors, alkaline phosphatase [13] Variable (increase, decrease, no change) changes in aminotransferases [13] Gastric pH increase [18] Decrease absorption of Ca^2+^ and Fe^2+^ [18]
Endocrine system	Glucose level increase (1–2 mg/dl each decade from 30 y.o.) Postprandial glucose concentration increase (4 mg/dl each decade from 30 y.o.) [12] Decrease in thyroid hormones, renin, aldosterone, growth hormone, testosterone, estrogens, vitamin D, and calciferol absorption [10,11,14,19,20] Increase in antidiuretic hormone
Immune system	Decreases in immunoglobulin G, immunoglobulin M, and bone marrow reserve (changes within the normal range) [15,16] Increases in antibody levels, immunoglobulin A, and erythrocyte sedimentation Rate [15,16] Rheumatoid factor false positive presence [17] Changes in lymphocyte number and function (e.g., tuberculin test false negative) [16]
Urinary tract	Decrease in glomerular filtration rate and creatine clearance (10 mL/min/1.73m^2^ each decade from 40 y.o.) [9]

**Table 2 metabolites-15-00479-t002:** Antioxidant enzymes in clinical investigations [77,78,79,80,81,82,83].

Enzyme of Interest	Study Population	Age Group	Disease/Condition of Interest	Results/Sample Material	Interpretation/Conclusion	Study No
CAT	Patients with MetS—long-term follow-up	55–75 y.o.	Metabolic syndrome	↑ CAT activity (no BMI reduction)/plasma	Compensatory response to persistent oxidative stress	[80]
CAT	Postmenopausal women with RA	48–64 y.o.	Rheumatoid arthritis	↑ CAT activity/serum	Anti-inflammatory effect of intermittent fasting	[81]
GPX	ACS patients—nutritional intervention	48–66 y.o.	Post-acute coronary syndrome	↑ GPX activity/serum	Reversible changes under redox-optimized diet	[79]
GPX, SOD	Patients with type 2 diabetes and CVD	~60 y.o.	Metabolic and cardiovascular comorbidities	↓ GPX/plasma and ↓SOD/RBC	Indicative of increased oxidative burden	[77]
PON1	General elderly population (PolSenior study)	≥65 y.o.	Age, metabolic dysfunction, inflammation	↓ Arylesterase activity/serum	Decline with age and inflammatory markers	[82]
PON1	Patients with CAD and type 2 diabetes	56–72 y.o.	Coronary artery disease	↓ PON1 activity/serum	Low PON1 linked to increased cardiovascular risk	[83]
SOD	General elderly cohort	≥65 y.o. (median of 86 years)	Mortality follow-up	↑ SOD activity in women/plasma	Associated with lower all-cause mortality	[78]

ACS—acute coronary syndrome; BMI—body mass index; CAD—coronary artery disease; CAT—catalase; CVD—cardiovascular disease; GPx—glutathione peroxidase; MetS—metabolic syndrome; PON1—paraoxonase 1; SOD—superoxide dismutase.

**Table 3 metabolites-15-00479-t003:** Oxidative stress genes linked to age-related diseases [43,79,88,89,90,91,92,93,94,95,96,97,98,99,100,101,102,103,104,105,106,107,108,109,110].

Gene	MIM Number	Type	Gene Function Result	Age-Related Disease	References
*CAT*	115500	Antioxidant	Decomposes hydrogen peroxide into water and oxygen	Cardiovascular disease, diabetes, cancer	[79,88]
*GPX1*	138320	Antioxidant	Reduces hydrogen peroxide to water	Atherosclerosis, diabetes	[89]
*GPX2*	603749	Antioxidant	Reduces hydrogen peroxide and lipid hydroperoxides	Cancer	[90]
*GPX4*	138322	Antioxidant	Reduces lipid peroxides, crucial for ferroptosis regulation	Cardiovascular disease, neurogenerative diseases, cancer	[91]
*GSTT1*	600436	Antioxidant	Detoxifies xenobiotics	Diabetes, cancer	[92]
*NOS1*	163731	Antioxidant	Produces nitric oxide, regulates neurotransmission	Neurogenerative diseases, hypertension	[93]
*NOS3*	163729	Antioxidant	Endothelial nitric oxide production, regulates vascular tone	Cardiovascular disease, hypertension	[93]
*NRF2*	600492	Antioxidant	Regulates antioxidant response elements (AREs)	Neurogenerative diseases, chronic obstructive pulmonary disease	[94]
*PON1*	168820	Antioxidant	Hydrolyzes lipid peroxides, anti-atherosclerotic	Cardiovascular disease, neurogenerative diseases	[95]
*PON2*	602447	Antioxidant	Cellular antioxidant, protects against oxidative stress	Cancer, neurogenerative diseases	[96]
*PON3*	602448	Antioxidant	Prevents LDL oxidation, anti-atherosclerotic	Atherosclerosis, metabolic syndrome	[96]
*PPARG*	601487	Antioxidant	Regulates lipid metabolism, inflammation control	Cardiovascular disease, diabetes	[97,98]
*SOD1*	147450	Antioxidant	Converts superoxide radicals into oxygen and hydrogen peroxide	Amyotrophic lateral sclerosis, Alzheimer’s disease, cardiovascular diseases	[43,99]
*SOD2*	147460	Antioxidant	Mitochondrial superoxide scavenging	Parkinson’s disease, cardiovascular diseases, cancer	[43,99]
*SOD3*	185490	Antioxidant	Extracellular superoxide scavenging	Inflammatory diseases, atherosclerosis	[43,100]
*ALOX15*	603693	Pro-oxidant	Produces lipid peroxidation products	Atherosclerosis, diabetes	[101]
*CYBA*	608508	Pro-oxidant	Component of NADPH oxidase, generates reactive oxygen species	Atherosclerosis, hypertension	[102]
*CYP1A1*	108330	Pro-oxidant	Metabolizes xenobiotics, generates oxidative metabolites	Cardiovascular disease, cancer	[97,103]
*MAO*	309850	Pro-oxidant	Catalyzes oxidation of neurotransmitters, produces hydrogen peroxide	Neurodegenerative diseases, Parkinson’s disease	[104]
*NOS2*	163730	Pro-oxidant	Produces nitric oxide in immune response	Inflammatory diseases, cancer	[105]
*NOX1*	300763	Pro-oxidant	Produces reactive oxygen species	Atherosclerosis, cancer	[106]
*NOX2*	300481	Pro-oxidant	Mediates oxidative burst in immune cells	Alzheimer’s disease, stroke	[107]
*NOX4*	605261	Pro-oxidant	Regulates reactive oxygen species in mitochondria	Cardiovascular disease, hypertension	[108]
*P66Shc*	600619	Pro-oxidant	Regulates mitochondrial reactive oxygen species production	Diabetes	[109]
*PTGS2*	600262	Pro-oxidant	Involved in prostaglandin synthesis, inflammation	Inflammatory diseases, cancer	[110]

*CAT*—gene encoding catalase; *GPX1*—gene encoding glutathione peroxidase; *GPX2*—gene encoding glutathione peroxidase 2; *GPX4*—gene encoding glutathione peroxidase 4; *GSTT1*—gene encoding glutathione S-transferase theta 1; *NOS1*—gene encoding nitric oxide synthase 1; *NOS3*—gene encoding nitric oxide synthase 3; *NRF2*—gene encoding nuclear factor erythroid 2-related factor 2; *PON1*—gene encoding paraoxonase 1; *PON2*—gene encoding paraoxonase 2; *PON3*—gene encoding paraoxonase 3; *PPARG*—gene encoding peroxisome proliferator-activated receptor gamma; *SOD1*—gene encoding superoxide dismutase 1; *SOD2*—gene encoding superoxide dismutase 2; *SOD3*—gene encoding superoxide dismutase 3; *ALOX15*—gene encoding arachidonate 15-lipoxygenase; *CYBA*— gene encoding cytochrome b-245 alpha subunit; *CYP1A1*—gene encoding cytochrome P450 1A1; *MAO*—gene encoding monoamine oxidase; *NOS2*—gene encoding nitric oxide synthase 2; *NOX1*—gene encoding NADPH oxidase 1; *NOX2*—gene encoding NADPH oxidase 2; *NOX4*—gene encoding NADPH oxidase 4; *P66Shc*—gene encoding Shc adaptor protein; *PTGS2*—gene encoding prostaglandin-endoperoxide synthase 2.

## Data Availability

No new data were created or analyzed in this study.

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
