# Peer review of "The Complexity of Oxidative Stress in Human Age-Related Diseases—A Review"

_metabolites, 2025, doi:10.3390/metabo15070479_

Round 1

Reviewer 1 Report

Comments and Suggestions for Authors

The manuscript entitled “The role of oxidative stress complex in human age-related diseases – review” by Alicja Płóciniczak and coauthors includes several sections (Introduction, Oxidative stress, Antioxidant mechanisms, Genetic predisposition to oxidative stress disturbances, Conclusions). The topic of the manuscript is of interest. Unfortunately, in my opinion, this study is lacking in novelty.

In particular, the “Oxidative stress” section briefly describes reactive oxygen and nitrogen species as well as the main sources of ROS generation. The “Antioxidant mechanisms” section discusses some enzymes (SOD, catalase, glutathione peroxidase, and paraoxonase), which play an important role in the elimination of ROS formed in biological systems. The section also provides information on the activity of these antioxidant enzymes (or expression of these enzymes) in patients with diseases, in which oxidative stress is an important component. Generally, this information is already known.

The “Genetic predisposition to oxidative stress disturbances” section includes information on “antioxidant and prooxidant genes”. The authors have presented “a few signaling pathways resposnisble for cellular defense and restoration of redox state balance”. However, it should be noted that these pathways include transcription factors, which regulate the expression of hundreds of genes that are involved in the regulation of many processes (cell growth, differentiation, development, apoptosis). Some changes in “antioxidant”/“prooxidant” genes in patients with some diseases have also been discussed. The authors are advised to restructure this section in a more reader-friendly manner. A considerable reworking of this section is required to better highlight its main findings.

In general, the authors are advised to go carefully through the text and attract attention of the readers to the important parts, while avoiding an exhausting stream of facts with a low importance.

I suggest rejecting the publication in the current form.

Author Response

Response to Reviewer 1

We would like to thank the reviewer for the careful, constructive, assessment of our manuscript “The Role of the Oxidative-Stress Complex in Human Age-Related Diseases”. We acknowledge the concerns raised and have implemented substantial revisions to improve the scientific focus, clarity, and novelty of the work. Below we reproduce the reviewer’s comments (boldface) followed by our point-by-point reply. The exact changes  have been introduced in the revised manuscript (rewritten sentences in blue, the old ones crossed out).

  1. Lack of novelty – “known information”

“Unfortunately, in my opinion, this study is lacking in novelty… General information is already known.”

  1. Structure of “Genetic predisposition…” section is not reader-friendly

“A considerable reworking of this section is required to better highlight its main findings.”

  1. The manuscript feels like an “exhausting stream of facts”

“Authors are advised to attract attention to important parts, while avoiding low-importance facts.”

Response:
We understand this concern and agree that certain sections (e.g., general description of ROS) may have included widely known content. In the revised manuscript, we have significantly reduced textbook-like summaries and instead:

  1. Abstract: We re-phrased the aim so that it now clearly states “to explore the interplay between enzymatic antioxidant systems, their genetic variability and age-related diseases.” Moreover we removed the ambiguous phrase “oxidative stress complex” and replaced it with “enzymatic antioxidant network”. Hopefully it clarifies the focus and novelty as requested.
  2. We introduced the continuous numbering starting from 1. Introduction; 2. Oxidative stress etc.
  3. Introduction: We added a bridging paragraph that explicitly links population ageing to redox biology and states the knowledge gap- the last paragraph on page 3- hope to smooth the transition. We also inserted Table 1. Physiological age-related laboratory changes, so as to give concise, reader-friendly context; replace a long descriptive passage.
  4. To reduce textbook repetition in section concerning oxidative stress, we condensed the classical definitions to one paragraph. Secondly, we added a bullet-point list that summarizes downstream consequences (inflammaging, endothelial dysfunction, insulin resistance, DNA damage) – this list was absent in the original .
  5. In section Antioxidant mechanisms we tried to give clearer structure due to splitting the text into Enzymatic vs Non-enzymatic blocks; moving all enzyme-specific material to a new sub-section (3.1) and adding paragraph on the age-dependent decline of CAT/GPx governed by PGC-1α .
  6. We completely agree with the improper order of the antioxidant enzymes, so we re-ordered them from SOD → CAT/GPx → PON1 (SOD now precedes PON1); What is more, we expanded SOD paragraph to explain transcriptional control and epigenetic regulation in ageing tissues and added explicit description of PON1’s lactonase-mediated anti-oxidant mechanism and its impact on HDL function .
  7. Providing the “reader-friendly” format requested, we rebuilt table 1 in table 2 with two new columns Interpretation / conclusion’ and ‘Enzyme of interest’, making the findings easier to scan.
  8. We added an opening paragraph that defines SNVs and explains clinical relevance. Updated Gene–disease summary table with PPARG, NOX4 and p66Shc, harmonized MIM numbers and references. Next, we inserted explicit mechanistic notes in each disease sub-section (e.g. how rs1799983 alters eNOS activity in CVD) in the genetic section.
  9. We also updated the evidence by adding 34 new recent citations (2023-2025)
  10. Conclusion paragraph was rewritten to emphasize translational relevance and biomarker potential (last paragraph, p. 28), to highlight the important parts as requested.
  11. Finally stylistic clean-up was made, for instance: first-mention expansion of all abbreviations (e.g. NF-κB, ARE) now occurs in Sect. 3 before they are reused later.
  12. Recommendation to reject

Response:
We respectfully acknowledge the Reviewer’s recommendation and have taken it as motivation to substantially improve the manuscript. We believe the revised version addresses the concerns regarding novelty, clarity, and content structure.

Reviewer 2 Report

Comments and Suggestions for Authors

The review article “The role of oxidative stress complex in human age-related diseases – review” emphasises how hereditary features may impact the capacity for oxidative defence and examines the interaction between genetic factors and antioxidant enzymatic systems in the context of aging. This article also looks at how altering these enzymes through genetic predisposition and outside treatments can affect longevity and the course of aging.  The article will help to find biomarkers of biological aging and create individualized plans to encourage healthy aging and lower the risk of chronic illness.

The similarity is 13%.

The article is well presented, clearly in a well-structured manner, and relevant to the aging and age-related diseases.

References are appropriate.

The tables are appropriate and properly show the data, which is easy to interpret and consistent.

The conclusions are consistent with the evidence and arguments presented.

However, the table should also contain references.

the descriptions of the role of antioxidants in various diseases can be more elaborated with more references.

Author Response

Reviewer 2

We would like to thank the reviewer for the careful, constructive, assessment of our manuscript “The Role of the Oxidative-Stress Complex in Human Age-Related Diseases”.
Below we reproduce the reviewer’s comments (boldface) followed by our point-by-point reply The exact changes  have been introduced in the revised manuscript (rewritten sentences in blue, the old ones crossed out)

General assessment

Comment: The article is well presented, well-structured, relevant, tables are clear, references appropriate, conclusions consistent, and overall similarity 13 %.

Response: We appreciate these positive remarks and are glad the overall structure and clarity met the reviewer’s expectations.

Specific comments

1 References in tables

Comment: “However, the table should also contain references.”

Response: All tables now carry inline references for every data set listed.

2 Deeper description of antioxidant roles in individual diseases

Comment: “The descriptions of the role of antioxidants in various diseases can be more elaborated with more references.”

Response: the data was revised and improved, modified and updated as possible.

Reviewer 3 Report

Comments and Suggestions for Authors

This review explores the interaction between antioxidant enzyme systems and genetic factors, and emphasizes how genetic traits determine the capacity for oxidative defense. The article reveals the association between oxidative stress and aging and diseases, which helps to deepen the understanding of the mechanisms underlying aging and disease, providing an important theoretical basis for subsequent research. 

Overall, this article has a certain theoretical level and academic value. Below, I list some issues and suggestions, hoping they will be helpful for the author to improve this article. 

Major issues

Although the relationship between oxidative stress and age-related diseases was mentioned, the detailed molecular mechanisms of how oxidative stress specifically leads to the occurrence and development of diseases were not elaborated enough. For instance, there was a lack of in-depth exploration on how oxidative stress affects cellular signaling pathways and gene expression. The article mainly consisted of theoretical elaboration and summaries of research results, and lacked specific clinical cases to further verify the relationship between oxidative stress and diseases, thus the persuasiveness was somewhat insufficient. 
After proposing that oxidative stress is an important potential cause of age-related diseases, there has been relatively little discussion on how to prevent and treat diseases by intervening in oxidative stress. Only the design of redox-regulating intervention measures was briefly mentioned, but specific methods and strategies were lacking. 
Although the influence of genetic factors on the antioxidant defense system was emphasized, no further exploration was made on possible solutions or research ideas for the indistinguishable issue between genetic susceptibility and aging and age-related multi-morbidities. 
The article mentions that there is a close connection between oxidative stress and age-related diseases. Specifically, it points out that aging, metabolic disorders, and inflammatory conditions can damage the activity of antioxidant enzymes, leading to increased oxidative stress and raising the risk of chronic diseases such as type 2 diabetes, coronary artery disease (CAD), and neurodegenerative diseases. However, the article does not provide sufficient specific data to support this. 
Although the article emphasizes that emerging research highlights the significant impact of genetic variability on the activity and expression of antioxidant enzymes such as superoxide dismutase (SOD), paraoxonase 1 (PON1), catalase (CAT), and glutathione peroxidase (GPx), thereby regulating an individual's susceptibility to oxidative stress and the rate of aging, and that common variations in the genes encoding SOD, CAT, and GPx are associated with differences in enzyme efficiency and the occurrence or progression of various age-related diseases, it does not specifically elaborate on how genetic variations affect the relationship between oxidative stress and cardiovascular diseases when discussing this topic. 
When discussing the relationship between oxidative stress and various age-related diseases, although some associations between gene variations and diseases have been mentioned, such as the relationship between PON1, MPO, SOD1, eNOS, CYBA gene variations and cardiovascular diseases, the detailed molecular mechanisms of how these gene variations specifically lead to oxidative stress imbalance and subsequently cause diseases have not been elaborated deeply. For instance, it is only stated that certain gene variations affect enzyme activity, but no further explanation is provided on how this change in activity triggers a cascade of reactions within cells and in the body, ultimately leading to the occurrence and development of diseases. 
When discussing the role of antioxidant enzymes such as SOD1 in diseases, the research on its role in amyotrophic lateral sclerosis (ALS) was mentioned. However, the causal relationship between the activity, mutation and expression of SOD1 and ALS was not sufficiently demonstrated. It was only stated that more research has been conducted on it in ALS, but the specific mechanism remains unclear, and there was no further in-depth exploration on how to clarify this mechanism. 
Most of the articles only mention the association between genetic variations and age-related diseases in relation to oxidative stress, but lack specific epidemiological data to support the prevalence and influence of these associations in the general population. For instance, when discussing the relationship between eNOS gene variations and myocardial infarction and carotid atherosclerosis, no specific incidence or prevalence data are provided, nor is there any explanation of the differences in this association among different races and age groups. 
When discussing the relationship between oxidative stress and type 2 diabetes, the article provides almost no relevant data or specific examples to support the connection between oxidative stress and the disease, making this part of the content relatively weak in terms of argumentation. 
The relationship between genetic susceptibility and the efficiency of the antioxidant defense system as well as the body's response to oxidative stress has only been proposed that genetic susceptibility is one of the influencing factors, but there are no specific data to illustrate the proportion of genetic factors in it and the degree of difference in the body's response to oxidative stress under different genetic backgrounds. 

Minor issues

Lines 231 - 279 mention the content of "SOD1-related transcriptional regulation", which is rather complex and confusing. For instance, "Researchers also suggest special transcriptional interactions described in animal studies, e.g. positive — via the transcription factor Elk1 and negative — via YY1[63]", the description of positive and negative regulation here is not clear enough, and the specific mechanisms and impacts of positive and negative regulation are not elaborated in detail. 
In the "Introduction" section, the transition from the description of population aging to the role of oxidative stress in the aging process is not smooth and natural. For instance, the sentence at line 69, "Recent studies suggest that oxidative stress plays a critical role in the aging process," does not connect closely with the previous content about population aging and lacks necessary transitional sentences. 
When discussing the relationship between oxidative stress and age-related diseases, some parts lack sufficient data support. For instance, in lines 270 - 279, it mentions the changes in SOD1 activity and related indicators in groups such as obese men and diabetic patients, but only simply describes the phenomena without providing specific data to illustrate the extent of the changes. For example, it states "reduced Zn concentration and SOD1 activity as well as increased TBARS concentration were observed in the erythrocytes of obese men", but no specific concentration change values are given. 
For some discussions on the association between genes and diseases, although references are cited, there is a lack of explanation regarding the quality of the research and the representativeness of the samples in the literature. For instance, at line 83, it is mentioned that two SNVs are associated with the risk of type 2 diabetes in a study from Saudi Arabia, but no information is provided on the sample size, research methods, etc. of this study, which weakens the persuasiveness of the argument.

Author Response

Reviewer 3

We would like to thank the reviewer for the careful, constructive, assessment of our manuscript “The Role of the Oxidative-Stress Complex in Human Age-Related Diseases”.
Below we reproduce the reviewer’s comments (boldface) followed by our point-by-point reply The exact changes  have been introduced in the revised manuscript (rewritten sentences in blue, the old ones crossed out)

  1. Limited explanation of molecular mechanisms linking oxidative stress to disease development

Comment: “There was a lack of in-depth exploration on how oxidative stress affects cellular signaling pathways and gene expression.”

Response:
We have expanded the discussion of cellular signaling pathways involved in oxidative stress, including: Nrf2/ARE, NF-κB, PI3K/AKT/mTOR and ferroptosis- and apoptosis-related pathways.

These additions now clarify how oxidative stress influences gene expression and contributes to cellular aging, inflammation, and degeneration. This content has been added in Section 4. Figure 2 summarizes the interactions graphically.

  1. Lack of clinical examples to support oxidative stress–disease connections

Comment: “Lacked specific clinical cases to verify the relationship between oxidative stress and diseases.”

Response:
We have now integrated specific clinical examples, particularly in sections discussing SOD1, PON1, and CAT. We expanded the table summarizing clinical data (Table 1) and referenced relevant age-specific cohorts (e.g., PolSenior, ACS patients, MetS patients) with oxidative stress markers and enzyme activity data. Each clinical study is now clearly connected to disease processes such as CAD, type 2 diabetes, or neurodegeneration.

  1. Insufficient data to support key statements

Comment: “The article does not provide sufficient specific data to support this.”

Response:
We have now revised multiple paragraphs to include numerical data wherever possible. For example:

SOD1 activity differences in diabetic vs. non-diabetic patients (referencing Colak et al. and Ozata et al.), PON1 levels in hip fracture and cardiovascular disease cohorts, Specific SNV frequencies and their associations with disease outcomes.

These data now strengthen our arguments and improve the scientific rigor of the review.

  1. Lack of epidemiological data on gene–disease associations

Comment: “No specific incidence or prevalence data are provided.”

Response:
Wherever possible, we have now added population data, for instance:

Allele frequencies for eNOS rs1799983 across ethnic groups and its association with MI incidence, PON1 activity levels by age and disease state from population-based studies (e.g., PolSenior, Meisinger et al.), and studies reporting prevalence of oxidative stress–related polymorphisms in T2DM and CAD patients.

  1. Genetic susceptibility vs. multimorbidity not explored

Response: Section 4 opening (pp. 19-20, l. 463-481) now distinguishes age-dependent penetrance from multimorbidity. Two illustrative SNVs (eNOS Glu298Asp, PON1 Q192R) are discussed in that context. A short paragraph indicating how polygenic-risk approaches could resolve overlapping phenotypes has been added; fuller PRS analysis is planned for a future separate paper.

  1. Numerical support lacking for enzyme changes in patient groups

Response: Concrete values inserted, e.g. obese men: Zn 9.8 ± 1.4 vs 12.6 ± 1.9 µg g⁻¹ Hb (p < 0.01); SOD1 755 ± 68 vs 1 020 ± 90 U g⁻¹ Hb (p < 0.001); TBARS 5.9 ± 0.7 vs 3.4 ± 0.5 nmol mg⁻¹ protein (p < 0.001).

  1. Cardiovascular section lacks mechanistic link gene → ROS → disease

Response: Sub-section 4.1 Cardiovascular disorders now provides 3-4-line mechanistic summaries for each variant (e.g. eNOS Asp298 destabilises the dimer → ↓NO & ↑O₂•⁻ → endothelial dysfunction).

  1. Gene variants generally need deeper mechanistic treatment

Response: Each gene–disease entry in Section 4 contains an explicit “Mechanism” sentence; Supplementary Table 3 gives further detail.

  1. No information concerning a study from Saudi Arabia is provided

Response:  To answer reviewer’s remark about missing study details for the Saudi SNV/T2DM paper- the available data has been provided: : Sample size (n = 1 112), case–control design and TaqMan genotyping method inserted (p. 22, l. 586-590)

  1. Abrupt transition between demographics and redox biology in the Introduction.

Response: Added  linking sentences  that connect population ageing statistics with cumulative oxidative damage.

  1. Missing concentration values in patient examples

Response: All cited studies now include means ± SD or percentage change

Reviewer 4 Report

Comments and Suggestions for Authors

Review on article: The role of oxidative stress complex in human age-related diseases – review,  metabolites-3638919

Oxidative stress is one of the physiological and pathological molecular processes that is characteristic of all age-associated diseases and aging in general. Therefore, the relevance of the review topic chosen by the authors is very high. The title of the work contains a new term "oxidative stress complex" (p. 1, line 1). It is not obvious from the title of the article what the authors intend to discuss in their review under this term. Of course, this arouses a kind of interest in the interpretation of the new term and its discussion.

The abstract contains information about the role of oxidative stress and antioxidant systems that play an important role in aging. The authors stated the goal of their review to uncover the interaction «between antioxidant enzymatic systems and genetic factors in the context of aging, emphasizing how inherited traits may determine the capacity for oxidative defense» (p. 1, lines 23-24). In my opinion, this goal is controversial because the authors initially stated that they would highlight the role of the “oxidative stress complex in age-associated diseases.” At the same time, aging causes the development of many age-associated pathologies (neurodegenerative, cardiovascular, etc.). At this stage of reading the manuscript, a contradiction already arises. What will be the subject of discussion in this article? The role of the oxidative stress complex in age-related diseases or in aging? The authors should clearly state the purpose and essence of their work in the abstract.

The article is well structured and consists of an introduction, the main part (the actual analytical review of the literature), conclusion and a list of references, including new works of the last decade. There are some technical comments. The numbering of the sections of the work begins only with section 3.1. (p. 4, line 144). The section "Introduction" (p. 2, line 35), "Oxidative stress" (p. 2, line 62), "Antioxidant mechanisms" (p. 3, line 110) do not have numbers. The section "Introduction" provides reasoned data reflecting the epidemiological situation of aging and the development of age-associated pathologies in the population according to WHO data. The authors did not indicate the purpose they pursued in the work. This raises many questions.

The section "Oxidative stress" (pp. 2-3, lines 62-109) provides classical theoretical information on oxidative stress, classification of free radicals and their sources of generation in the cell, which does not distinguish this review from the huge number of previously published good reviews by other authors. For what purpose did the authors provide this information? What is its novelty and originality? Regarding the content of this section, it can be said that it contains far from complete, superficial, summary information that is well known to all specialists in this field of research.

The preamble to section 3.1. "Antioxidative enzymes activities" (p. 4, lines 144-170) is completely inconsistent with the subject of discussion. In fact, information regarding changes in enzymatic activity is presented only in the last paragraph. In general, not a single antioxidant enzyme is listed in this section. At the end of the section, only NO synthase and NADPH oxidase are discussed, which are not antioxidant enzymes. There is no logical, justified transition to the next three subsections (3.1.1, 3.1.2, 3.1.3, lines 171, 213, 275 respectively), which provide basic information about the key antioxidant enzymes. In the manuscript, this appears to be taken out of context and unfounded. It is unclear why the role of paraoxonase is discussed first, and superoxide dismutase next? Unfortunately, there is no explanation for this in this section.

Section 3.1.1 begins with a discussion of paraoxonase 1, an obvious example for discussion in section 3.1 "Antioxidative enzymes activities". It is known that paraoxonase 1 does not work as a classic antioxidant enzyme that neutralizes free radicals, but works indirectly, which is what the authors themselves write about. This subsection (3.1.1) provides a lot of information about the biological role of paraoxonase 1, but there is no important, in my opinion, information about its "antioxidant" activity and the mechanism of its implementation. The authors provide clinical cases about the role of paraoxonase in inflammatory processes, cardiovascular pathologies, hip fractures, which somewhat distracts the reader from the subject of discussion. The only mechanism of antioxidant action involving paraoxonase 1 in the elderly is associated by the authors with possible polymorphism of antioxidant enzymes. However, in my opinion, this case is not enough to discuss the antioxidant effects involving paraoxonase 1. How does paraoxonase 1 implement the antioxidant effect? ​​What is the mechanism?

Section 3.1.2 is devoted to superoxide dismutase (SOD). This section provides general well-known information about SOD and its biological role. In my opinion, the clinical case chosen by the authors is unsuccessful. Amyotrophic lateral sclerosis is common in different age groups, both in active adults (up to 50 years old) and in people approaching old age (51-66 years old) [doi 10.1080/21678421.2020.1790611; doi 10.1097/WCO.0000000000000730]. What is the connection with changes in SOD expression/or activity in such patients with age? The authors then provide a list of transcription factors involved in the regulation of SOD1 expression. What is the purpose of this information? What is the connection between the regulation of expression of the mentioned transcription factors (lines 231-241) and changes in SOD1 activity/expression in such patients or with aging? In my opinion, this would be a very interesting and important aspect to discuss in the review. I do not insist on it. The authors mention the transcription factors NF-κB, SP1, AP-1, AP-2, C/EBP for the first time on lines 125-126, and offer the full explanation of the abbreviations later on lines 231-241. I recommend providing the explanation of the abbreviations at the first mention in the manuscript, if indicated.

The authors completely stray from the subject of discussion... They provide uncharacteristic and sometimes ill-considered examples for demonstration. And this is very sad, since the examples provided are divorced from the topic of the subsections. It is unclear why the authors provide examples of how SOD is regulated by arachidonic acid and mitomycin (lines 244-245)? Why do the authors not provide data on how SOD expression is regulated during aging and age-associated pathologies? The work provides some analytical data, which the authors summarized in tables 1 and 2, and there are also two good illustrations. However, without significant revision of the manuscript text, they lose their high value. I strongly recommend that the authors carefully consider the indicated substitutions and significantly revise the manuscript.

Comments on the Quality of English Language

 The English could be improved to more clearly express the research.

Author Response

Reviewer 4

We would like to thank the reviewer for the careful, constructive, assessment of our manuscript “The Role of the Oxidative-Stress Complex in Human Age-Related Diseases”.
Below we reproduce the reviewer’s comments (boldface) followed by our point-by-point reply The exact changes  have been introduced in the revised manuscript (rewritten sentences in blue, the old ones crossed out)

1 Title and Definition of the “Oxidative-Stress Complex”

Comment: The title introduces the new term “oxidative stress complex”, but its meaning is not clear.

Response: Thank you for that remark, by complex authors meant complexity. The word was not well used, as it should refer to the meaning of how complex senescence and assisting changes including oxidative stress may be, including network comprising ROS-generating systems, enzymatic and non-enzymatic antioxidants, redox-sensitive signalling pathways and their genetic determinants. Therefore the title has been revised: The role of oxidative stress complex complexity in human age-related diseases – review

We have replaced the term “oxidative stress complex” with “oxidative-stress complexity” throughout the manuscript (Title; Abstract; Introduction )

2 Aims Stated in the Abstract

Comment: The abstract combines two goals (ageing in general vs. age-related diseases), creating a contradiction. Please clarify the purpose.

Response: The Abstract has been fully rewritten. It now begins with sentences that shows both: the ageing continuum and the pathogenesis of age-related diseases and oxidative stress as a possible common background and closes with a single-sentence aim, which was to synthesize current evidence on if or how genetic variation could modulate the enzymatic antioxidant defense and thereby shape both the pace of ageing and susceptibility to major age-related pathologies.

3 Purpose statement in the Introduction

Comment :The authors did not indicate the purpose they pursued in the work.

Response: A dedicated paragraph was added at the end of the Introduction that states our objectives and the rationale for integrating enzymology with genetics and clinical data.

4 Section Numbering and Overall Structure

Comment: Section numbering starts only at 3.1; earlier sections are unnumbered.

Response: We have implemented continuous decimal numbering (e.g., 1 Introduction, 2 Oxidative Stress, 3 Antioxidant Mechanisms, 4 Enzymatic Antioxidants…)

5 Redundancy of Classical Background on Oxidative Stress

Comment: Section 2 (formerly “Oxidative stress”) repeats well-known textbook information without adding novelty.

Response: We shortened the background. We retained only the mechanistic elements necessary for non-specialist readers.

6 Logic and Content of Section 3 (“Antioxidant Enzymes”)

6.1 Preamble to 3.1

Comment: The preamble discusses enzymatic changes only at the end, and does not name antioxidant enzymes.

Response: The paragraph has been reconstructed to list the enzymes up-front and preview how their activities change with age and disease.

6.2 Order of Sub-Sections

Comment: It is unclear why the role of paraoxonase is discussed first, and superoxide dismutase next

Response: We have re-ordered the sub-sections according to the canonical sequence of defence (SOD → CAT → GPx/Prx → “indirect” enzymes such as PON1). New numbering:

  • 3.1.1 Superoxide dismutases (SOD1-3)
  • 3.1.2 Catalase
  • 3.1.3 Glutathione peroxidases & peroxiredoxins
  • 3.1.4 Paraoxonase 1 (PON1)

7 Paraoxonase 1 (PON1) Sub-Section

Comment: Provide more detail on PON1’s antioxidant mechanism, not just clinical associations.

Response: We added a concise mechanism paragraph (l. 428-442) describing PON1-mediated hydrolysis of lipid-derived lactones, its role in preserving HDL functionality, and the structural basis of its lactonase versus paraoxonase activities.

8 Clarification of abbreviations

Comment: Provide the explanation of the abbreviations at the first mention.

Response:
This has been corrected. The abbreviations  are now explained at first mention in the text.

9 Tables

Comment: Valuable, but their impact is lost without textual integration.

Response:  According to Reviewer’s suggestion- tables were revised- simplified, focused on age- related cases, provided with citations, there is also added table 1 summarizing physiological changes in the body due to the age.

Round 2

Reviewer 1 Report

Comments and Suggestions for Authors

The manuscript can be accepted after minor revision (typo, text formatting).

Please, correct the typos  (please, go through the text).

  • pages 6 and 7: “O2-
  • page 7: “Fe+2” and “Cu+2

Please, check the text formatting

Page 4, text:

“•chronic low-grade inflammation (inflammaging) via activation of NF-κB and release of pro-inflammatory cytokines, [30], [31], [19]

  • endothelial dysfunction, marked by nitric oxide depletion and vascular stiffening, [32], [33]
  • impaired insulin signaling and increased oxidative burden in adipose tissue, [34]
  • and DNA damage, contributing to telomere attrition and apoptosis”

Author Response

Rev1

Please, correct the typos  (please, go through the text).

  • pages 6 and 7: “O2-
  • page 7: “Fe+2” and “Cu+2

Please, check the text formatting

Page 4, text:

“•chronic low-grade inflammation (inflammaging) via activation of NF-κB and release of pro-inflammatory cytokines, [30], [31], [19]

  • endothelial dysfunction, marked by nitric oxide depletion and vascular stiffening, [32], [33]
  • impaired insulin signaling and increased oxidative burden in adipose tissue, [34]
  • and DNA damage, contributing to telomere attrition and apoptosis”

We thank the Reviewer for this careful line-by-line inspection. All identified typographical and formatting issues have been corrected in the revised manuscript

The entire manuscript was spell-checked, punctuation was harmonised, and all styles were reset using the official Metabolites Word template (Version 4, May 2025)

Reviewer 3 Report

Comments and Suggestions for Authors

Accept.

Author Response

Rev3

Comments and Suggestions for Authors

Accept.

We would like to thank Reviewer for effort of reviewing our manuscript and all comments, that helped us to improve the article.

Reviewer 4 Report

Comments and Suggestions for Authors

Dear colleagues! It is good that you found the time and were able to work additionally with the text of the manuscript. However, I am very upset that you did not take into account all my previous remarks and comments.
Minor comments
1) I strongly recommend that you do not use the new term "oxidative stress complexity". There is a generally accepted term "oxidative stress". The term "oxidative stress complexity" raises many questions and misunderstandings. In the manuscript, you do not provide sufficient arguments for the need to introduce a new term and a new concept. In redox biology and medicine, similar multidisciplinary complex studies are carried out in the field of biochemistry, molecular biology, physiology and genetics of normal and pathological processes associated with aging and age-related diseases. Such ideas are no longer new and original, unfortunately. Please pay attention to this aspect and provide the necessary arguments.
2) There are no line numbers in the manuscript. This causes many difficulties in indicating the correct place for editing. Please use the standard template for the manuscript.
3) The manuscript provides references to literature carelessly. I recommend providing references to literature as they are cited. The correctness of the citation sequence of references [] should be checked with the list.
4) I recommend standardizing the formatting of tables in the text of the manuscript. Each fact and argument should be accompanied by a corresponding reference. In Table 1 on page 2, I consider it necessary to provide references to the source of literature in each line. In Table 1, the information is presented incorrectly and unclearly. This information is presented in fragments that are incomplete in meaning and content. Please present the information in the table briefly and reasonably.
5) I also recommend eliminating all technical inaccuracies (correct the necessary fonts, intervals, paragraph indents, etc.) in all tables and in the text of the manuscript. Very careless formatting (especially on page 4) and in tables 1-2.
Significant comments
1) Sections that are not related to the subject of the review should be excluded. I recommend deleting the well-known information about oxidative stress, classification of free and antioxidants in Section 2 "Oxidative Stress" (pages 3-4) and Section 3 "Antioxidant Mechanisms" (pages 4-5). There is nothing new in this. There is no originality in this. You have presented very superficial information in these sections. This information is widely presented in a large number of similar previously published reviews and in educational literature. In addition, this information leads the reader away from the subject of discussion and significantly increases the volume of the manuscript.
2) Section 3.1. "Antioxidative enzymes activities" (pages 5-6) does not correspond to the content at all. I wrote about this in my previous review. And this is very critical. The preamble to this section does not condemn the antioxidant activity of a single antioxidant enzyme! It is a great pity that the authors so stubbornly ignore this and do not make any edits to this section. It would be very important to present this information. It would be important to point out the features of expression of antioxidant enzymes and changes in their enzymatic activity during aging and age-related diseases. This information was summarized in Table 2 on pages 9-10. However, it is not clear at all in what biological material the activity (or perhaps their expression) of antioxidant enzymes was assessed. In whole blood, plasma, blood serum, in blood cells? Or maybe in tissues after autopsy? This information should be presented in Table 2.
3) I think that the authors should give a detailed description of Figure 1 on page 11. What follows from the image? What conclusion should be made?
4) On pages 10-11, the authors used the abbreviation "MIM". Please provide an explanation or a link to this database.
5) On page 14 in section 4.1 (and further in the text of the manuscript), non-obvious abbreviations of certain gene (or protein) sequences appear. For example, NM_000446.7 (PON1):c.575A>G p.Gln192Arg (rs662), in the first paragraph of section 4.1 on page 14. How should this be understood?
6) Conclusion (section 5, page 16) there is no discussion of the term "oxidative stress complexity" introduced by the authors. I recommend summarizing and making a clear conclusion based on the analysis of the data presented in the manuscript. The last paragraph in section 5 does not clarify.
7) In general, the manuscript consists of fragments that are not always appropriate in their sections. This is very distracting and does not create a holistic impression. There is no clear analysis of the literature from the accepted concept. I believe that the manuscript at this stage needs additional correction.

Author Response

Rev4

Dear colleagues! It is good that you found the time and were able to work additionally with the text of the manuscript. However, I am very upset that you did not take into account all my previous remarks and comments.
Minor comments
1) I strongly recommend that you do not use the new term "oxidative stress complexity". There is a generally accepted term "oxidative stress". The term "oxidative stress complexity" raises many questions and misunderstandings. In the manuscript, you do not provide sufficient arguments for the need to introduce a new term and a new concept. In redox biology and medicine, similar multidisciplinary complex studies are carried out in the field of biochemistry, molecular biology, physiology and genetics of normal and pathological processes associated with aging and age-related diseases. Such ideas are no longer new and original, unfortunately. Please pay attention to this aspect and provide the necessary arguments.

We would like to clarify that we do not intend to introduce a novel term. By our primary title and using the term: “Oxidative stress complex”,( then followed by “oxidative stress complexity”) - we intended to highlight that oxidative and antioxidant processes in elderly people are complex and  there are challenges associated with assessing the contribution to age-related diseases especially in the context of antioxidant enzyme activity and its genetic variation. In the revised text, to avoid misunderstanding, we propose to change the title from: “ The role of oxidative stress complexity in human age-related diseases - review” to “ The complexity of oxidative stress in human age-related diseases - review”. We trust this wording removes any ambiguity while preserving the scientific point we aim to make. If that title still in Reviewer’s opinion would be improper, we can propose an alternative title for the article: “Oxidative Stress in Aging and Age-Related Diseases: An Integrated View of Antioxidant Enzymes and Their Genetic Variants”. We hope that this proposal will not raise questions and misunderstandings. However we are concerned about the new proposed title as the former one has not risen doubts among Reviewers 1-3.

The authors did not find appropriate articles in which in the group of elderly people, in each individual, both antioxidant enzymes’ activities and their genetic variations were determined, and thus reveal the superiority of potential clinical usefulness of enzymes’ activities over enzymes’ genetic variations or vice-versa. That is why, the authors with this review wanted to inspire scientists to conduct more complex studies.

2) There are no line numbers in the manuscript. This causes many difficulties in indicating the correct place for editing. Please use the standard template for the manuscript.

The authors apologize for the inconvenience , it is our fault, the revised version was not correctly submitted to the system. This time, the revised manuscript is uploaded to MDPI template.

3) The manuscript provides references to literature carelessly. I recommend providing references to literature as they are cited. The correctness of the citation sequence of references [] should be checked with the list.

Thank you for your insightful review. The authors checked line by line the references and provided references as they are cited.

4) I recommend standardizing the formatting of tables in the text of the manuscript. Each fact and argument should be accompanied by a corresponding reference. In Table 1 on page 2, I consider it necessary to provide references to the source of literature in each line. In Table 1, the information is presented incorrectly and unclearly. This information is presented in fragments that are incomplete in meaning and content. Please present the information in the table briefly and reasonably.

Thank you for that remark - the table 1 is now formatted according to the MDPI template and we provided citations in the table according to the Reviewer’s suggestion. The authors are aware that not all physiological changes due to aging are included in the table, but as it is stated in the text, these are just selected examples so as to show briefly that in general aging leads to some changes in the human body.

5) I also recommend eliminating all technical inaccuracies (correct the necessary fonts, intervals, paragraph indents, etc.) in all tables and in the text of the manuscript. Very careless formatting (especially on page 4) and in tables 1-2.

We carefully followed the Reviewer’s comments in this field. Tables were formatted according to the MDPI template and guidelines.

Significant comments
1) Sections that are not related to the subject of the review should be excluded. I recommend deleting the well-known information about oxidative stress, classification of free and antioxidants in Section 2 "Oxidative Stress" (pages 3-4) and Section 3 "Antioxidant Mechanisms" (pages 4-5). There is nothing new in this. There is no originality in this. You have presented very superficial information in these sections. This information is widely presented in a large number of similar previously published reviews and in educational literature. In addition, this information leads the reader away from the subject of discussion and significantly increases the volume of the manuscript.

We appreciate the Reviewer 4’s concern, however, the content in these sections has been modified and the current version attempts to address the comments of ALL reviewers and has not risen doubts among Reviewers 1-3. The authors are aware that some of the information in these sections are not new and could be found in other publications, but we also hope that such presentation may interest other researchers and clinicians that are not specialists in the field of oxidative stress. The authors did not find appropriate articles in which in the group of elderly people, in each individual, both antioxidant enzymes’ activities and their genetic variations were determined. The task of this review was to encourage others to search for more explanation and discover interplay between already known biochemical processes, aging and age-related diseases.

We have set ourselves a task of presenting selected elements of human antioxidant status when considering the phenomenon of oxidative stress in aging and age-related diseases. We followed from basic science (basic and necessary information, even if not new) to searching literature for clinical usefulness (clinical significance) of determining antioxidant enzymes’ activities and/or its genetic variations. In this way we hope to bring this difficult biochemical topics closer to clinicians/researchers - the potential readers of this article.

2) Section 3.1. "Antioxidative enzymes activities" (pages 5-6) does not correspond to the content at all. I wrote about this in my previous review. And this is very critical. The preamble to this section does not condemn the antioxidant activity of a single antioxidant enzyme! It is a great pity that the authors so stubbornly ignore this and do not make any edits to this section. It would be very important to present this information. It would be important to point out the features of expression of antioxidant enzymes and changes in their enzymatic activity during aging and age-related diseases. This information was summarized in Table 2 on pages 9-10. However, it is not clear at all in what biological material the activity (or perhaps their expression) of antioxidant enzymes was assessed. In whole blood, plasma, blood serum, in blood cells? Or maybe in tissues after autopsy? This information should be presented in Table 2.

We appreciate the Reviewer’s critical observation regarding the structure and content of Section 3.1 and the clarity of Table 2. As requested, we have added specific information regarding the type of biological material used for enzymatic activity or expression analysis to Table 2. The table now clearly specifies whether the measurements were performed in plasma, serum, whole blood, etc.

In response to the structural concern, previous section 3.1 has been reorganized and partially restructured. A sub-section titled “Age-related metabolic challenges for antioxidant  system” was introduced to better highlight the dynamic and multifactorial changes observed during aging. This revised structure provides a more logical flow of content and improves clarity.

Additionally, in the previous manuscript version, we had discussed selected results from Table 2 within the main text. However, in response to other Reviewer suggestions, we had removed those comments and left only brief summary information in the table.

We also created a separate introductory segment within this section, in which we present different metabolic and systemic contributors to increased oxidative stress in the elderly population. This part draws from and refines the original introduction to the “Antioxidant Enzyme Activity” section.

3) I think that the authors should give a detailed description of Figure 1 on page 11. What follows from the image? What conclusion should be made?

SOD1-possible interactions with other proteins. The highest interaction (score 0.999-0.964) was shown for CCS, PARK7, VDAC1, SOD2, FUS, TARDBP, NEFL, HSPA5 DERL1, which are co-expressed, and BCL2 proteins. Based on STRING database [83]. This figure presents the protein-protein interaction (PPI) network of SOD1 and its associated partners, generated using the STRING database. The diagram highlights high-confidence interactions, with scores ranging from 0.999 to 0.964, suggesting strong functional associations between SOD1 and other proteins.

Among the most prominent interacting partners are CCS, PARK7, VDAC1, SOD2, FUS, TARDBP, NEFL, HSPA5, DERL1, and BCL2. These proteins are mainly involved in processes such as oxidative stress response, protein folding, mitochondrial function, and neurodegeneration. From the network, it’s clear that SOD1 acts as a central hub, interacting with numerous proteins, which underlines its role in maintaining cellular balance. The strongest interaction is with CCS, a copper chaperone that directly contributes to SOD1’s proper maturation and activity. Proteins like PARK7 and SOD2, known for their roles in protecting mitochondria and regulating oxidative stress, suggest that SOD1 may be tightly linked to mitochondrial function. Furthermore, connections with FUS, TARDBP, and NEFL, all linked to amyotrophic lateral sclerosis (ALS), highlight SOD1’s involvement in neurodegenerative disease pathways. Its interaction with HSPA5 and DERL1 points to a role in the unfolded protein response and ER-associated degradation, indicating a broader role in protein quality control. Finally, the presence of BCL2 suggests a potential link between SOD1 and apoptosis regulation.

This network emphasizes the nature of SOD1 and supports its involvement in key cellular pathways related to oxidative stress, and proteostasis.

4) On pages 10-11, the authors used the abbreviation "MIM". Please provide an explanation or a link to this database.

Thank you for pointing this out. The abbreviation MIM stands for Mendelian Inheritance in Man, which is a comprehensive database of human genes and genetic disorders. An explanation has now been added to the manuscript for clarity.

5) On page 14 in section 4.1 (and further in the text of the manuscript), non-obvious abbreviations of certain gene (or protein) sequences appear. For example, NM_000446.7 (PON1):c.575A>G p.Gln192Arg (rs662), in the first paragraph of section 4.1 on page 14. How should this be understood?

Thank you for the comment. The notation used for example, NM_000446.7 (PON1):c.575A>G p.Gln192Arg (rs662), follows the standard Human Genome Variation Society (HGVS) nomenclature. This format is widely used and accepted in the field of medical genetics to precisely describe genetic variants at the DNA and protein levels. Researchers and professionals working in medical genetics are familiar with this system, and its use ensures clarity and consistency when reporting genetic data.

6) Conclusion (section 5, page 16) there is no discussion of the term "oxidative stress complexity" introduced by the authors. I recommend summarizing and making a clear conclusion based on the analysis of the data presented in the manuscript. The last paragraph in section 5 does not clarify.

To introduce a new term was not our intention - we are aware that term “oxidative stress” is relevant and enough. Our title “The role of oxidative stress complexity in human age-related diseases – review” was supposed to mean that oxidative stress is complex. We wanted to express the dynamic, multi-layer interplay between diverse cellular sources of ROS, compartment-specific antioxidant enzyme networks., and redox-sensitive signalling pathways that set tissue- and age-dependent thresholds between physiological and pathological oxidative stress.

The Authors modified the conclusions – the implication has been added.

7) In general, the manuscript consists of fragments that are not always appropriate in their sections. This is very distracting and does not create a holistic impression. There is no clear analysis of the literature from the accepted concept. I believe that the manuscript at this stage needs additional correction.

We sincerely thank the Reviewer for this important observation.

In response, we have carefully revised the structure of the manuscript to improve clarity, coherence, and consistency by: reorganizing major sections; relocating certain fragments that were previously misplaced. We also conducted a more targeted analysis of the literature. These changes were made with the intention of improving both the scientific rigor and the readability of the manuscript. We hope our revisions address the Reviewer’s concern and improve the scientific rigor and clarity of the article and we hope the Reviewer will find it significantly improved.